# An RNA-seq based comparative approach reveals the transcriptome-wide interplay between 3′-to-5′ exoRNases and RNase Y

Laura Broglia [1,2,3,5], Anne-Laure Lécrivain [1,2,4,5], Thibaud T. Renault [1,2,3], Karin Hahnke[1,2], Rina Ahmed-Begrich[1,2], Anaïs Le Rhun [1,2 ✉] & Emmanuelle Charpentier [1,2,3,4 ✉]

RNA degradation is an essential process that allows bacteria to control gene expression and adapt to various environmental conditions. It is usually initiated by endoribonucleases (endoRNases), which produce intermediate fragments that are subsequently degraded by exoribonucleases (exoRNases). However, global studies of the coordinated action of these enzymes are lacking. Here, we compare the targetome of endoRNase Y with the targetomes of 3′-to-5′ exoRNases from *Streptococcus pyogenes*, namely, PNPase, YhaM, and RNase R. We observe that RNase Y preferentially cleaves after guanosine, generating substrate RNAs for the 3′-to-5′ exoRNases. We demonstrate that RNase Y processing is followed by trimming of the newly generated 3′ ends by PNPase and YhaM. Conversely, the RNA 5′ ends produced by RNase Y are rarely further trimmed. Our strategy enables the identification of processing events that are otherwise undetectable. Importantly, this approach allows investigation of the intricate interplay between endo- and exoRNases on a genome-wide scale.

[1] Max Planck Unit for the Science of Pathogens, D-10117 Berlin, Germany. [2] Max Planck Institute for Infection Biology, Department of Regulation in Infection Biology, D-10117 Berlin, Germany. [3] Institute for Biology, Humboldt University, D-10115 Berlin, Germany. [4] The Laboratory for Molecular Infection Medicine Sweden (MIMS), Umeå Centre for Microbial Research (UCMR), Department of Molecular Biology, Umeå University, S-90187 Umeå, Sweden. [5]These authors contributed equally: Laura Broglia, Anne-Laure Lécrivain. ✉email: anais.le-rhun@inserm.fr; research-charpentier@mpusp.mpg.de

The ability to modulate gene expression enables bacteria to rapidly adapt to their environment. Ribonucleases (RNases) regulate transcript abundance, leading to RNA maturation (e.g., for tRNAs, rRNAs), stabilization or degradation. Eventually, all transcripts—even the most stable—are degraded by RNases, leading to the renewal of the nucleotide pool.

As a general rule, RNA degradation starts with an endonucleolytic processing in the RNA body, leading to the generation of decay intermediates. Those are further digested by exoRNases and, finally, by oligoRNase/nanoRNases[1]. The main endoRNases that have been demonstrated to initiate RNA decay are RNase E in Gram-negative bacteria and its functional orthologue RNase Y in Gram-positive bacteria[2,3]. However, in many Gram-positive bacteria, RNA degradation can also be initiated by the complex of RNases J1/J2, which displays both endo- and 5′-to-3′ exoribonucleolytic activities[4–6]. The initial processing of a transcript is the limiting step of the RNA decay and the access of endoRNases to transcripts is usually restricted. For instance, RNase E favours 5′ monophosphorylated (5′ P) transcripts and cleaves 2 nt upstream of a uridine (U) in A/U rich regions[7,8]. RNase Y also prefers 5′ P transcripts[9] and additional requirements have been described depending on the orthologue studied. In *Staphylococcus aureus*, RNase Y processes transcripts preferably downstream of a guanosine (G)[10]. In *Streptococcus pyogenes*, a G is required for the in vivo processing of the *speB* transcript, encoding a major virulence factor[11]. In *Bacillus subtilis* and *S. aureus*, RNase Y processing relies on proximal RNA secondary structures[9,12].

The decay intermediates, once generated by endoRNase(s), are cleared immediately from the cell by 3′-to-5′ exoRNases[13]. In *E. coli*, the decay intermediates are mainly degraded by the 3′-to-5′ exoRNases II, R and PNPase[14,15]. In *B. subtilis* and *S. pyogenes*, the major 3′-to-5′ exoRNase is PNPase[16,17]. In addition, in *B. subtilis*, the 3′-to-5′ exoRNases PH and YhaM participate in RNA decay, albeit with lower efficiency than the main 3′-to-5′ exoRNase[16]. *S. pyogenes* YhaM exhibits a very short processivity (3 nt on average) on a large number of RNA 3′ ends, the impact of which on mRNA decay is currently unknown[17].

With the emergence of RNA sequencing techniques allowing the global detection of RNase cleavage sites, several targetomes of endoRNases have been determined, such as those of RNase Y in *S. aureus* and *B. subtilis*[10,18]. To date, the activity and specificity of 3′-to-5′ exoRNases towards decay intermediates produced by a given endoRNase have never been studied on a global scale.

Here, we present a comparative RNA-seq based approach that allows us to dissect the complex landscape of RNA ends in *S. pyogenes*. We study the interplay of endoRNase Y with 3′-to-5′ exoRNases in *S. pyogenes*, a pathogen causing a wide range of diseases in humans. We determine the first targetome of RNase Y in this bacterium and compare it with three 3′-to-5′ exoRNase targetomes, previously characterized by our laboratory[17]. We show on a global scale that RNase Y mainly acts in concert with PNPase during RNA degradation. In this regard, we demonstrate a role of the RNase Y-PNPase interplay in the control of the differential stability of polycistronic mRNAs and the decay of 5′ regulatory elements. This strategy allows us to elucidate the interplay and dynamics of endoRNase- and exoRNase-mediated RNA processing events otherwise not detectable when RNases are studied separately.

## Results

**In vivo RNase Y targetome**. To identify RNase Y processing positions, we compared the abundance of RNA ends (5′ and 3′) in the *S. pyogenes* wild type (WT), RNase Y deletion mutant (Δ*rny*) and complemented RNase Y deletion mutant (Δ*rny::rny*) strains, as described previously for other RNases in this bacterium[17,19]

(Fig. 1a and Methods). A total of 320 RNA ends, which were more abundant in the WT than in the Δ*rny* strain, were retrieved: 190 RNA 5′ ends and 130 RNA 3′ ends (Fig. 1b and Supplementary Data 1). Because these ends depended on the presence of RNase Y, we referred to these positions as "*rny*_ends". We could not identify 5′ and 3′ *rny*_ends located at neighbouring nucleotides, indicative of a single processing event (Fig. 1c). Therefore, we deduced that the upstream and/or downstream RNA fragments generated by RNase Y processing are degraded by exoRNases. When several ends were identified at consecutive nucleotides, only one position was kept (see Methods) and named "stepped" (S). When ends mapped to one nucleotide, they were referred to as "unique" (U) positions (Fig. 1d). The 5′ and 3′ *rny*_ends harboured distinct features. First, the 3′ ends were mainly S-RNA ends whereas the identified 5′ ends were mostly U-RNA ends (Fig. 1e). Second, the RNA 5′ and 3′ ends differed in the nature of the sequence found in their proximity. Indeed, we observed the presence of a G upstream of the RNA 5′ ends (87.4% of the cases), which we did not observe for the RNA 3′ ends (Fig. 1f). This strong preference for G indicates that this nucleotide might play an important role in RNase Y target recognition and/or processing (Fig. 1f). Third, we observed a decrease in the minimum free energy (MFE) upstream of the 3′ *rny*_ends, indicative of a putative RNA structure. In contrast, the MFE in proximity of the 5′ *rny*_ends increased compared with the surrounding regions, indicating that these sequences corresponded to single-stranded RNA regions (Fig. 1f). The observation that the 3′ *rny*_ends identified by our analysis harboured a "stepped" profile and that the G was not conserved suggests that these ends could result from trimming by 3′-to-5′ exoRNases, which do not stop precisely at an exact nucleotide[20–22]. Because these 3′ *rny*_ends depend on the presence of RNase Y, trimming by 3′-to-5′ exoRNases would be subsequent to RNase Y processing.

**Comparison of the RNase Y and 3′-to-5′ exoRNase targetomes**. To investigate whether the 3′ *rny*_ends originated from 3′-to-5′ exoribonucleolytic activity, we compared the RNase Y targetome with the targetomes of three 3′-to-5′ exoRNases (PNPase, YhaM and RNase R) recently characterized by our laboratory[17] (Fig. 2). In our previous study, we identified the processing sites of 3′-to-5′ exoRNases by comparing the abundance of the RNA 3′ ends between the WT and 3′-to-5′ exoRNase mutant (Δ*exornase*) strains. The RNA 3′ ends more abundant in the Δ*exornase* strain and the 3′ ends more abundant in the WT strain were annotated as exoRNase trimming start and stop positions, respectively (Supplementary Fig. 1)[17]. The targetomes were compared using two different approaches, which are described in the next two sections, and we found that 58% of the identified 3′ *rny*_ends corresponded to 3′-to-5′ exoRNase—mainly PNPase—trimming start or stop positions (Fig. 2). We believe that the remaining 42% of the *rny*_3′ ends were also further trimmed upon processing by RNase Y, as we could not detect a preference for a G at these positions. We did not match these 3′ *rny*_ends with 3′-to-5′ exoRNase start or stop positions because they were likely targeted by several 3′-to-5′ exoRNases at once or by unidentified RNases. Overall, we conclude that PNPase is the main 3′-to-5′ exoRNase that acts in concert with RNase Y to degrade RNAs in *S. pyogenes* (Fig. 2).

**Pairing 3′ rny_ends and exoRNase trimming stop positions**. In our comparative analysis, we identified 46, 27 and one 3′ *rny*_ends corresponding to PNPase, YhaM and RNase R trimming stop positions, respectively, suggesting that the trimming of these RNAs was RNase Y dependent (Fig. 2 in red, Fig. 3a,

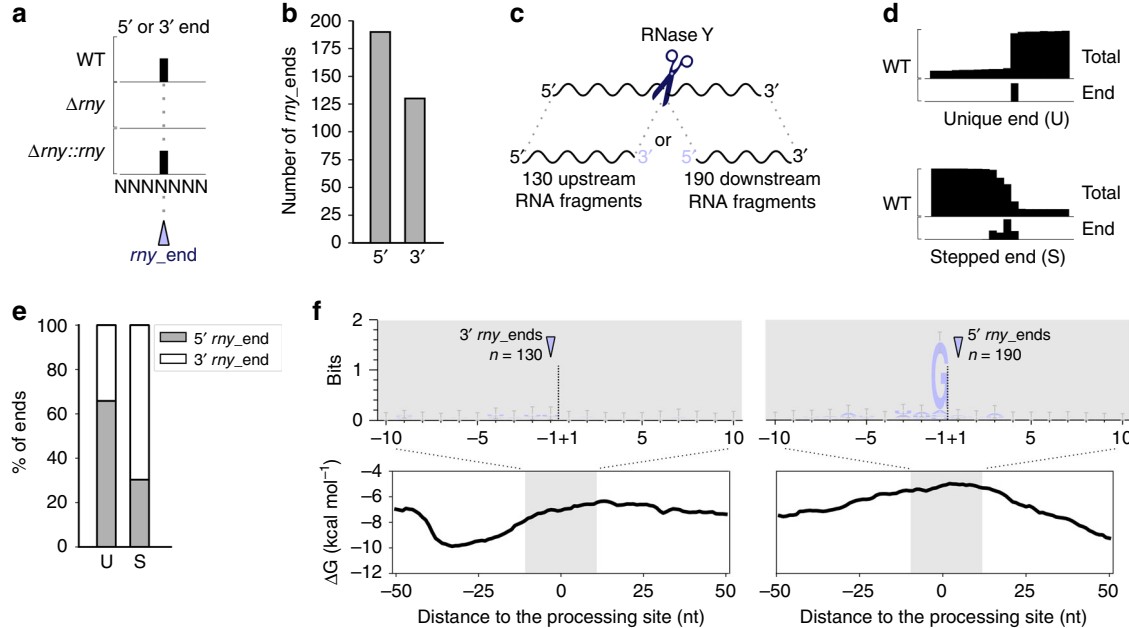

**Fig. 1 RNase Y processes RNAs after a guanosine. a** Representation of RNA end (5′ or 3′) profiling obtained by RNA sequencing (performed in biological triplicates). The RNA ends that were more abundant in the wild type (WT) and complemented *rny* deletion strain (Δ*rny::rny*) than in the RNase Y deletion strain (Δ*rny*) are annotated as *rny*_end. "NNNNNNN" represents a sequence processed by RNase Y. **b** The bar plot shows the number of 5′ or 3′ ends that were more abundant in the WT than in the Δ*rny* strain (see Methods). **c** RNase Y cleavage (scissors) generates two processing products. We never retrieved both the RNA fragments upstream and downstream of the cleavage site for the same processing event. **d** Schematic drawing of total and end (5′ or 3′) coverages from RNA sequencing, illustrating RNA 5′ "unique" (U) and 3′ "stepped" (S) end positions. **e** Proportion of RNA 5′ and 3′ ends classified as U and S. **f** Sequence and structure conservation of the identified 5′ and 3′ *rny*_ends. The logo was created from the alignment of all sequences 10 nt on each side of the identified ends. Error bars are automatically calculated by the WebLogo library and correspond to an approximate Bayesian 95% confidence interval. The minimum free energy (ΔG) was calculated at each nucleotide position using a sliding window of 50 nt over the entire genome. The average ΔG (kcal mol$^{-1}$) calculated for a window of 100 nt centred on the identified ends is depicted.

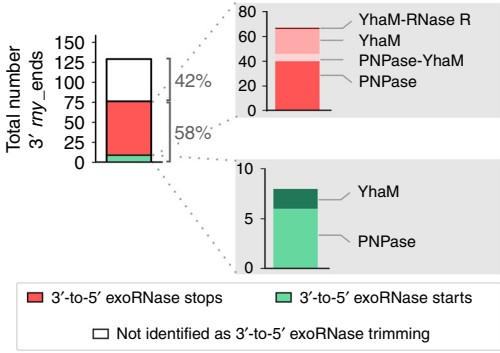

**Fig. 2 3′-to-5′ exoRNases trim RNAs generated by RNase Y processing.** Left bar plot: portion of RNA 3′ ends (3′ *rny*_ends) corresponding to 3′-to-5′ exoRNase start positions (bottom portion), 3′-to-5′ exoRNase stop positions (middle portion) and not associated with 3′-to-5′ exoRNase (top portion). Right bar plots: number of trimming starts (top) and stops (bottom) that correspond to 3′ *rny*_ends, which were uniquely produced by PNPase and YhaM or produced by two different 3′-to-5′ exoRNases.

Supplementary Data 2). With this comparison, we identified RNase Y-processed RNAs that were targeted by 3′-to-5′ exoRNases and not entirely digested (Fig. 3a). For a few examples, we observed that the RNAs were trimmed by YhaM and an additional 3′-to-5′ exoRNase: one RNA was trimmed by YhaM and RNase R, and six were trimmed by PNPase and YhaM (Fig. 2 and Supplementary Data 2).

We further aimed to identify the initial RNase Y processing position of these targets and we hypothesized that it would correspond to the position where the exoRNase starts trimming.

Therefore, we searched for 3′-to-5′ exoRNase trimming start positions that were located downstream of the 3′ *rny*_ends (Fig. 3a). We retrieved 19 and 5 trimming start positions for PNPase and YhaM, respectively, which could correspond to the RNase Y initial processing positions (Fig. 3a and Supplementary Data 3). We observed enrichment of G at the 19 PNPase trimming start positions (Fig. 3b), with 9 mapped to a G and 9 located 1 or 2 nt upstream of a G (Supplementary Data 3). Considering the frequency of G around the PNPase start positions, we hypothesize that the initial RNase Y processing (PNPase trimming start positions) actually occurs at G (Fig. 3a and b). The 1 or 2 nt distance from G is likely due to the known nibbling activity of YhaM that we observed in *S. pyogenes*[17]. Similarly, 4 trimming start positions of YhaM were located at a G (Supplementary Data 3). One other position, located at an adenosine, corresponded to a predicted PNPase trimming stop position and was identified in the transcript encoding the putative SPy_0316 protein (Supplementary Data 3). Upon RNase Y processing, the SPy_0316 mRNA was trimmed first by PNPase and then by YhaM (Fig. 3d). We indeed identified a 3′ *rny*_end corresponding to PNPase and YhaM stop positions (Supplementary Data 2, Fig. 3d). PNPase started trimming, 34 nt upstream of the stop position (Fig. 3d and Supplementary Data 5), at a G corresponding to the initial RNase Y processing position, followed by YhaM, which stopped at the base of a stem loop predicted in the middle of the SPy_0316 open reading frame (ORF) (Fig. 3e). The YhaM trimming start position was not detected in the absence of PNPase (Fig. 3d), which confirmed that YhaM targeted the RNA 3′ end generated by PNPase. Similarly, we observed subsequent trimming of PNPase and YhaM upon RNase Y processing in the intergenic region between Spy_sRNA73113 and *rplO*, encoding the 50S ribosomal

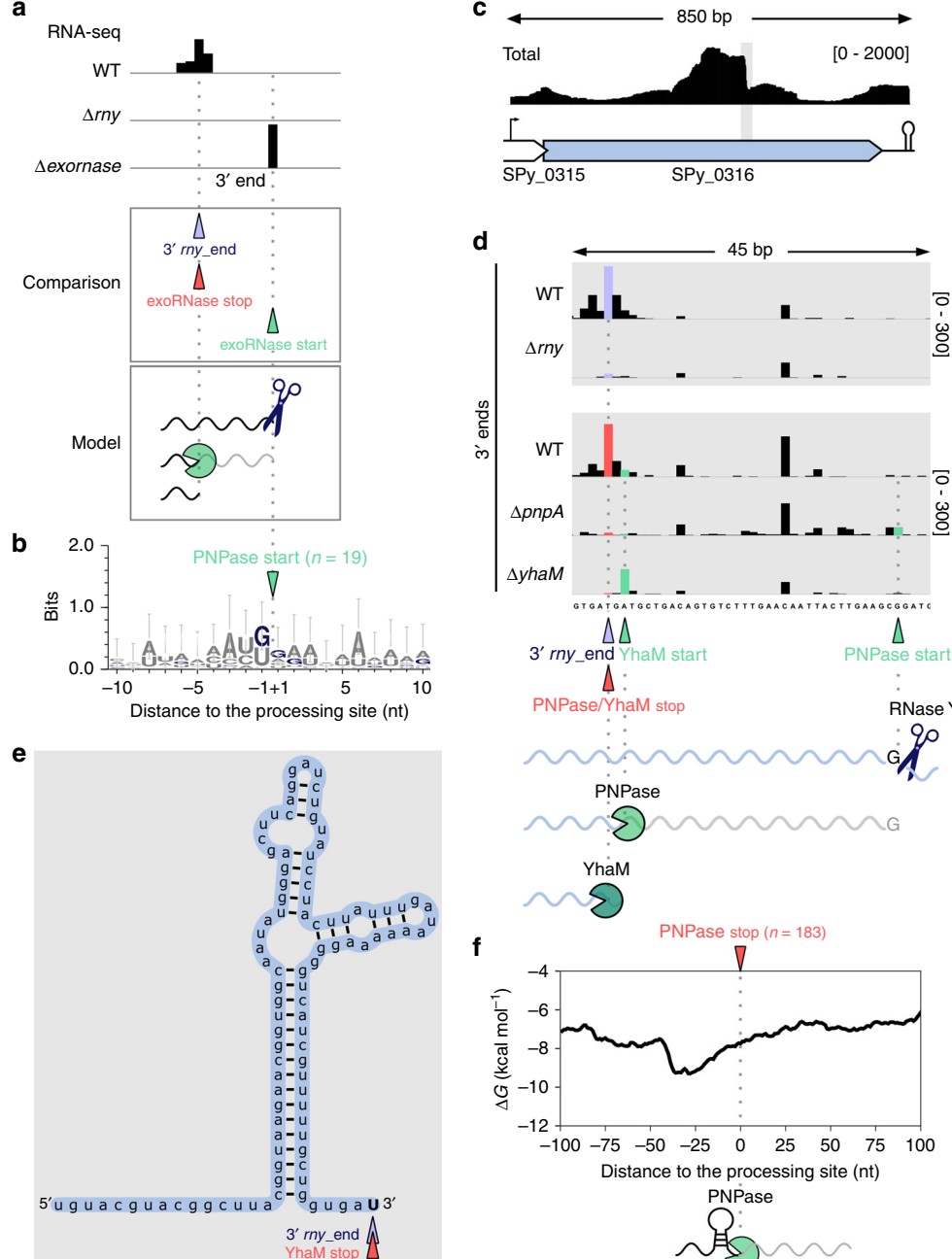

**Fig. 3 RNase Y-generated RNAs are degraded by PNPase until secondary structures are encountered. a** Upper panel: example of 3′ end coverage profiling from RNA sequencing. Middle panel: the RNA ends that were more abundant in the WT than in the Δ*rny* strain are indicated below the coverage and depicted with purple arrowheads (3′ *rny*_ends). The RNA ends corresponding to the trimming start and stop positions of exoRNases are depicted with green and red arrowheads, respectively (see Supplementary Fig. 1). The 3′ *rny*_ends were paired to 3′-to-5′ exoRNase stop positions (Supplementary Data 2 and Fig. 2, bottom). Bottom panel: the 3′-to-5′ exoRNases ('pacman' symbols) started trimming upon RNase Y (scissors) processing and stopped before the RNA termini. The 3′ *rny*_ends corresponding to the 3′-to-5′ exoRNase stop positions were compared with the exoRNase trimming start positions located downstream. The 3′-to-5′ exoRNase start position corresponds to the initial RNase Y processing position (Supplementary Data 3). **b** The logo, displaying the information (bits), was created from the alignment of all sequences surrounding the 19 identified PNPase trimming start positions. **c** Total coverage of SPy_0316 (encoding a putative transcriptional regulator) in WT obtained by RNA sequencing, and schematic representation of the locus. The grey rectangle indicates the region where the processing sites of RNase Y, PNPase and YhaM were identified. **d** 3′ end coverage of a portion of SPy_0316 in the WT, Δ*rny*, YhaM deletion mutant (Δ*yhaM*) and PNPase deletion mutant (Δ*pnpA*) strains. The coverage scales are indicated between brackets. RNase Y processed the RNA after a G, corresponding to the detected PNPase trimming start position. PNPase trimmed 34 nt of the SPy_0316 RNA 3′ end. This new RNA 3′ end was subsequently nibbled by YhaM. **e** RNA folding of the region 100 nt upstream of the 3′ *rny*_ends corresponding to YhaM trimming stop positions. YhaM started trimming after PNPase stopped, at the base of the stem loop structure, and consequently removed 2 nt from the RNA 3′ end. **f** Structure conservation at the 183 PNPase stop positions previously identified[17]. The decrease in the minimum free energy (ΔG, kcal mol⁻¹) is indicative of RNA structures likely preventing PNPase from degrading the RNA.

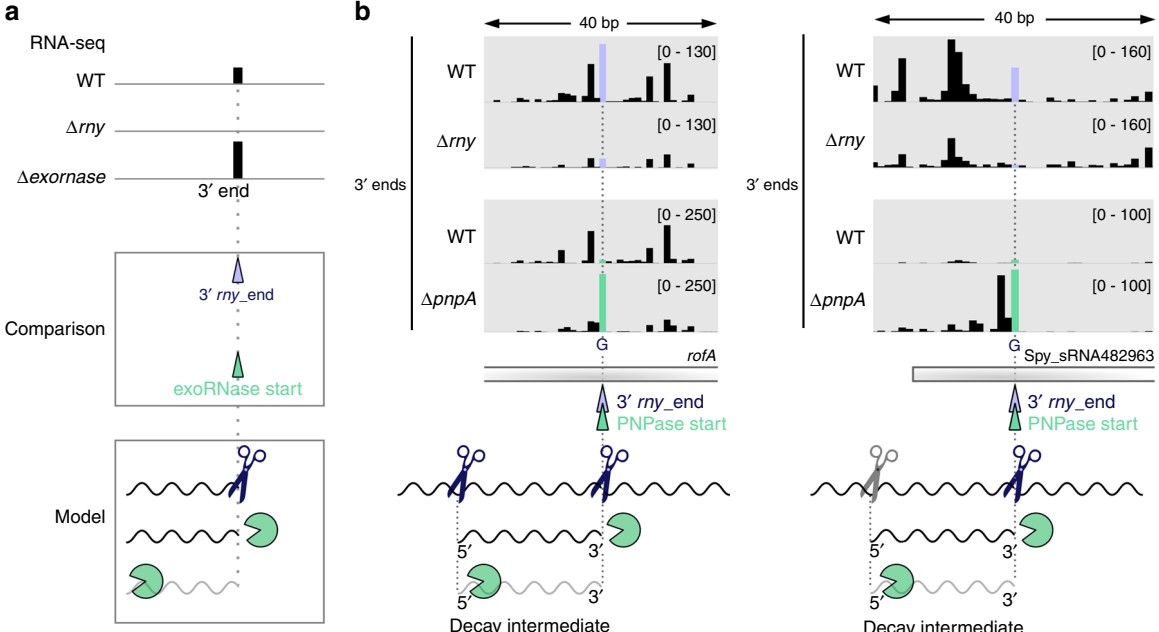

**Fig. 4 Initial RNase Y cleavage position validated by PNPase trimming start positions. a** Upper panel: example of 3′ end coverage profiling from RNA sequencing. Middle panel: the RNA ends that were more abundant in the WT than in the Δ*rny* strain are indicated below the coverage and depicted with purple arrowheads (3′ *rny*_ends). The RNA ends corresponding to the trimming start positions of exoRNases are depicted with green arrowheads (see also Supplementary Fig. 1). Bottom panel: The identified 3′ *rny*_ends were paired to 3′-to-5′ exoRNase start positions (Supplementary Data 4 and Fig. 2, top). The 3′-to-5′ exoRNase targeted the RNA fragments generated by RNase Y processing, but the exoRNase start position was nonetheless detectable in WT, indicating that a portion of the RNAs was not degraded. The 3′-to-5′ exoRNase start position corresponds to the initial RNase Y processing position (Supplementary Data 4). **b** Examples of RNAs identified by matching the 3′ *rny*_ends (purple arrowheads) with PNPase trimming start positions (green arrowheads) (see also Supplementary Fig. 3). For each RNA, the RNA 3′ end profile obtained from RNA sequencing in the WT, Δ*rny* and Δ*pnpA* strains is shown and the scales are indicated between brackets. The RNA 3′ ends generated by RNase Y (blue scissors) and eventually targeted by PNPase ('pacman' symbol) corresponded to decay intermediates. For *rofA* mRNA, RNase Y was also responsible for the generation of the 5′ end of the decay intermediate. For Spy_sRNA482963, another endoRNase (grey scissors) produced the 5′ end of the decay intermediate, which was previously identified as an RNA 5′ end that was more abundant in the Δ*pnpA* strain than in the WT strain[17].

protein L15 (Supplementary Fig. 2). The concerted action of these RNases is likely involved in the sRNA 3′ end production (Supplementary Fig. 2). These examples illustrate that in addition to targeting RNA 3′ ends after terminator regions and endoRNase processing[17], YhaM also trims RNA 3′ ends generated by other 3′-to-5′ exoRNases.

In the previous examples, we observed that PNPase stopped trimming these RNAs until encountering stem loop structures, suggesting that these structures prevented further degradation. In our previous publication, we calculated the average MFE (ΔG, in kcal mol⁻¹) around PNPase stop positions using 25 nt-long sequences, which are sufficient for detection of terminator structures, and we could not predict any structure[17]. Here, we used 50 nt sequences for the calculation, allowing us to detect variations in ΔG that are indicative of structures longer and weaker than the structures of terminator regions. Indeed, we observed a decrease in the MFE upstream of the PNPase stop positions (Fig. 3f). Therefore, we now conclude that PNPase can be blocked by RNA structures, as previously described in vitro for PNPase from other bacteria[21,23,24].

**Pairing 3′ rny_ends and exoRNase trimming start positions.** Six PNPase and two YhaM trimming start positions corresponded to the RNA 3′ ends produced by RNase Y (Fig. 2 in green and Fig. 4a). Therefore, the 3′ ends generated by RNase Y are targeted by these exoRNases. Their detection in this analysis suggests that a portion of the RNAs had not yet been subjected to 3′-to-5′ exoRNase degradation.

PNPase trimming start positions were located at a G and probably corresponded to the RNA 3′ ends generated by RNase Y (Fig. 4b and Supplementary Fig. 3). For the *rofA*, Spy_sRNA482963, *ezrA* and *htrA* transcripts, PNPase trimming starts corresponded to the 3′ ends of previously identified decay intermediates (Fig. 4b and Supplementary Fig. 3), which were degraded by PNPase up to the 5′ end of the decay intermediate[17]. The two other targets, namely, *rpsU* and the intergenic region between Spy_sRNA1696464 and Spy_sRNA1696905, were also likely degraded up to the RNA termini (5′ ends), as the PNPase stop positions were not detected (Supplementary Fig. 3). Notably, it is known that PNPase usually releases 2- to 5-nt-long oligoribonucleotides, which are then further degraded by oligoRNase/nanoRNases[25]. For simplicity, in the following text, we write that PNPase degrades RNAs up to their termini.

**Pairing 5′ rny_ends to exoRNase trimming start positions.** We identified 190 RNA 5′ *rny*_ends that could not be paired with 3′ *rny*_ends, meaning that the RNA 3′ ends produced during the same processing events were not detected in the WT strain (Fig. 5a, Supplementary Data 1). In the comparisons described above, we observed that most of the 3′ *rny*_ends were targeted by 3′-to-5′ exoRNases. Therefore, 3′-to-5′ exoRNases most likely also degraded the RNA fragment upstream of the RNase Y processing positions, which explains why we were not able to detect those RNAs.

To investigate this hypothesis, we paired the 5′ *rny*_ends to 3′-to-5′ exoRNase trimming start positions. In particular, we screened for trimming start positions within 10 nt upstream of the 190 RNA

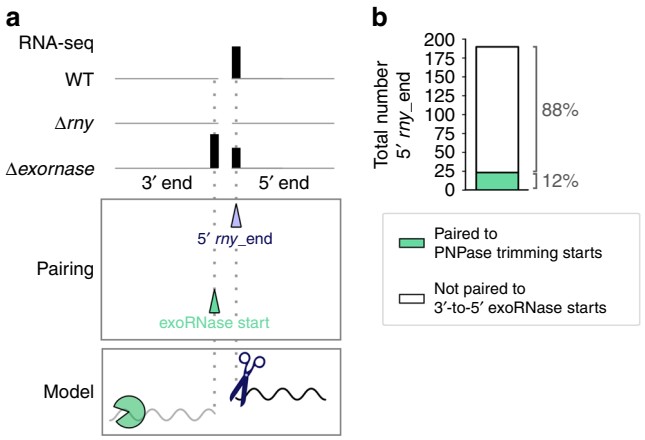

**Fig. 5 PNPase completely degrades the RNase Y-generated RNAs located upstream of the processing site. a** Example of 3′ and 5′ end coverage profiling by RNA sequencing. The RNA ends that were more abundant in the WT strain than in the Δ*rny* strain (purple arrowhead) and the RNA ends corresponding to trimming start positions of exoRNases (green arrowhead) are indicated below the coverage. The 5′ *rny*_ends were paired to 3′-to-5′ exoRNase start positions that were located at least 10 nt upstream (Supplementary Data 5). Subsequently to RNase Y activity, the 3′-to-5′ exoRNases completely degraded the RNA fragments upstream of the RNase Y processing. **b** Portion of RNA 5′ ends (5′ *rny*_ends) paired with 3′-to-5′ exoRNase start positions. All the trimming starts associated with 5′ *rny*_ends were PNPase trimming start positions.

5′ ends generated by RNase Y (Fig. 4a, Supplementary Data 5). We determined that 12% of the RNA 5′ ends were located in proximity of PNPase trimming start positions, indicating that the generated RNA fragment upstream of the processing site was degraded by this exoRNase (Fig. 5b; Supplementary Data 5). The observation that a majority of the PNPase trimming start positions were located up to 4 nt apart from the 5′ ends generated by RNase Y could be explained again by the activity of YhaM. The remaining 88% of the RNA 5′ ends were not associated with 3′-to-5′ exoRNase trimming start positions (Fig. 5b); therefore, the fate of the RNA fragment upstream of RNase Y processing could not be determined with our comparative analysis.

The identification of RNA 5′ and 3′ ends, the generation of which was RNase Y dependent, coupled with the comparison of exoRNase trimming start and stop positions, allowed us to provide an accurate and precise annotation of the RNase Y targetome. Overall, PNPase appears to be the major 3′-to-5′ exoRNase that degrades the RNA 3′ ends produced by RNase Y (Figs. 2 and 5). Interestingly, the PNPase-RNase Y double-deletion strain (Δ*pnpA*Δ*rny*) grew slower than both the Δ*rny* strain and the YhaM-RNase Y double-deletion strain (Δ*yhaM*Δ*rny*) (Supplementary Fig. 4), which indicates that RNase Y and PNPase genetically interact and play an important role in bacterial physiology.

**RNase Y produces short RNA fragments.** To identify fragments with both ends produced by RNase Y (two cleavages in the same RNA molecule) (Fig. 6a), we calculated the distance between the 5′ *rny*_ends and the 3′ *rny*_ends (Fig. 6b). We observed that, by setting a maximum distance of 1000 nt, in a majority of the cases, the 5′ *rny*_ends and the 3′ *rny*_ends were 50–200 nt apart (Fig. 6b, Supplementary Data 6). Examples of these fragments were indeed detectable in the WT, but not in the Δ*rny* strain, when examined by northern blot analyses (Fig. 6c and Supplementary Fig. 5). We further explored whether the 3′ *rny*_ends of these putative fragments were targeted by 3′-to-5′ exoRNases and

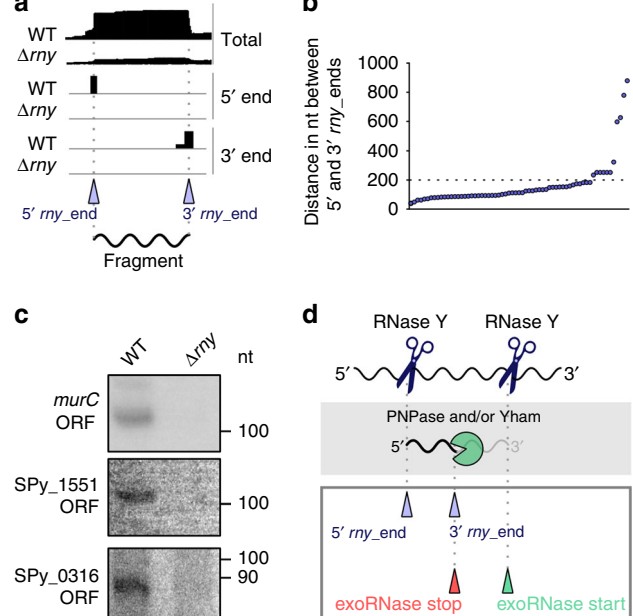

**Fig. 6 Characterization of RNase Y-generated RNA fragments in the WT strain. a** Representation of total, 5′ and 3′ end coverage profiles in the WT and Δ*rny* strains obtained by RNA sequencing and corresponding to RNA fragments produced by RNase Y. The 5′ and 3′ *rny*_ends are indicated with purple arrowheads. **b** The positions of the 5′ and 3′ *rny*_ends were compared by setting minimum and maximum distances of 40 and 1000 nt, respectively, between the ends. Each dot represents paired 5′ and 3′ *rny*_ends (Supplementary Data 6). **c** Northern blot analyses of RNA fragments in the open reading frames (ORFs) of *murC*, SPy_1551 and SPy_0316, generated by RNase Y and detectable only in the WT strain. The full blots, the loading controls and the RNA sequencing profile for each fragment are shown in Supplementary Fig. 5c–e, and the source data are provided as a Source Data file. Shown are the results of one representative northern blot analysis (*n* = 3). **d** Schematic representation of the generation of the short RNA fragments. RNase Y (blue scissors) is responsible for the production of both 5′ and 3′ fragment ends. The intermediate RNA fragment 3′ ends are, in 60% of the cases (Supplementary Data 6), subsequently trimmed by PNPase and/or YhaM ('pacman' symbol) from the start position (green arrowhead) until the stop position (red arrowhead).

noticed that 60% of them were trimmed by PNPase and/or YhaM (Supplementary Data 6, exemplified in Fig. 6d). The reason why these fragments were detectable (not degraded in the WT strain) remains unknown. We observed a decrease in the MFE at the fragment 3′ ends, indicating the presence of a stable structure (Supplementary Fig. 6a and b). Therefore, it is possible that the fragments were highly resistant to degradation because they were protected by this structure.

Among the RNA fragment 3′ ends generated by RNase Y, 23% were trimmed by YhaM. In some cases, we observed that the fragment present in the WT (Fig. 6c and Supplementary Fig. 5c–e) was not detected in the Δ*yhaM* strain, by neither northern blot analyses nor RNA sequencing (Supplementary Fig. 5c–e). We therefore wondered whether, in the absence of YhaM, these fragments were digested by PNPase or RNase R. However, in both the Δ*pnpA*Δ*yhaM* and Δ*rnr*Δ*yhaM* double-deletion strains, we did not detect the fragments by northern blot analyses (Supplementary Fig. 5c–e). It is possible that YhaM exerts a protective role by preventing further degradation of these fragments. Alternatively, the redundancy between RNase R and PNPase or the involvement

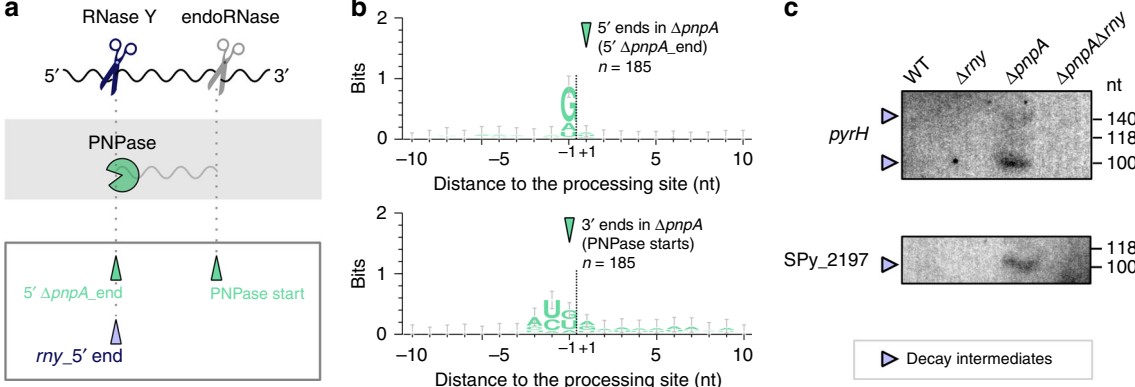

**Fig. 7 RNase Y produces decay intermediates degraded by PNPase. a** Schematic representation of decay intermediates generated by RNase Y (blue scissors) and rapidly degraded by PNPase ('pacman' symbol). RNase Y produced the decay intermediate 5′ end, which was identified as an RNA 5′ end that was more abundant in the Δ*pnpA* strain than in the WT strain. The decay intermediate 3′ end, corresponding to the PNPase trimming start position, was probably produced by an unidentified endoRNase (grey scissors). **b** Sequence conservation around the 5′ ends (5′ *pnpA*_ends) and 3′ ends (PNPase starts) of the decay intermediate from the 185 decay fragments previously identified, present in the Δ*pnpA* strain and not in the WT strain. **c** Northern blot analyses of decay intermediates (in *pyrH* and SPy_2197) in the WT, Δ*rny*, Δ*pnpA* and Δ*pnpA*Δ*rny* strains. The RNase Y-generated fragments degraded by PNPase (purple arrows) are indicated. Shown are the results of one representative northern blot analysis (n = 3). The full blots, loading controls, RNA sequencing-based RNA end profile with the detected 5′ and/or 3′ *rny*_ends, and trimming start positions are shown in Supplementary Fig. 6 and the source data are provided as a Source Data file.

of another RNase could explain the absence of the fragment in the Δ*pnpA*Δ*yhaM* and Δ*rnr*Δ*yhaM* strains.

**RNase Y produces decay intermediates degraded by PNPase.** As recently shown in *E. coli*, PNPase is actively involved in the degradation of small RNA fragments derived from transcripts targeted by sRNAs[26]. We previously observed that PNPase rapidly degraded decay intermediates produced by endoRNases in *S. pyogenes*[17]. The 5′ ends of these decay intermediates were identified as RNA ends that were more abundant in the Δ*pnpA* strain than in the WT strain (185 5′ Δ*pnpA*_ends)[17] (Fig. 7a). Here, we observed a conserved G located upstream of the decay intermediate 5′ ends that was not observed at the decay intermediate 3′ ends (Fig. 7b)[17]. Based on the RNase Y cleavage signature inferred from our analysis, we propose that the decay intermediates harbouring a G at the 5′ end (127 decay intermediates) were generated by RNase Y (Fig. 7a and b). Indeed, the decay intermediates, visualized by northern blot analyses, were detected in the Δ*pnpA* strain but not in the Δ*pnpA*Δ*rny* strain, indicating that RNase Y was involved in their production (Fig. 7c and Supplementary Fig. 6). The decay intermediate 3′ ends could result from RNase Y processing—followed by exoRNase trimming, explaining the lack of G conservation—or from processing by another endoRNase (Fig. 7a and Supplementary Fig. 6).

**Role of RNase Y and PNPase in the 5′ regulatory element degradation.** A portion of the decay intermediates degraded by PNPase are derived from endoRNase processing of regulatory RNA 5′ UTRs (e.g., T-boxes and riboswitches)[17]. Here, we observed that some of these decay intermediates were produced by RNase Y (Fig. 8 and Supplementary Fig. 7). For example, RNase Y processing generated decay intermediates from the *serS* and *thrS* T-box RNA 5′ UTRs, providing access for PNPase to digest these RNAs further up to the 5′ end (Fig. 8a and Supplementary Fig. 7). For all the regulatory elements analysed, the decay intermediates accumulated in the Δ*pnpA* strain and were not present in the Δ*pnpA*Δ*rny* strain, demonstrating that RNase Y is required for initiation of the decay of the premature terminated transcripts derived from the T-box and riboswitches (Fig. 8b and Supplementary Fig. 7).

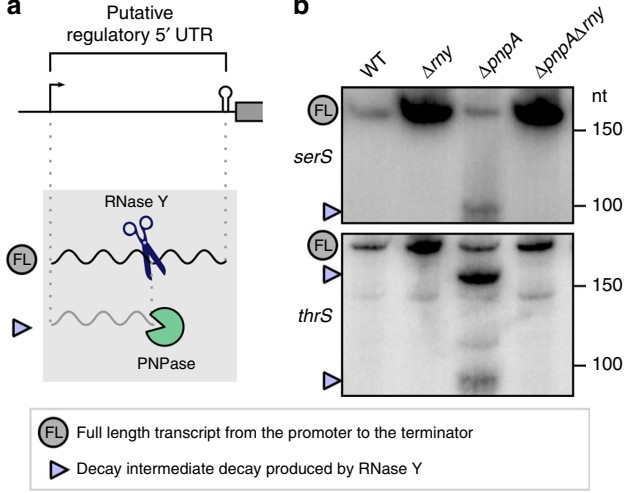

**Fig. 8 RNase Y initiates the exoRNase-mediated degradation of putative regulatory 5′ UTRs. a** Left, models of decay of putative regulatory elements. RNase Y (blue scissors) processes regulatory 5′ UTR elements, producing decay intermediates that are subsequently degraded by PNPase ('pacman' symbol) (see also Supplementary Fig. 7). **b** Northern blot analyses of putative T-boxes (*serS* and *thrS*) in the WT, Δ*rny*, Δ*pnpA* and Δ*pnpA*Δ*rny* strains. The full-length (FL) and RNase Y-generated decay intermediates (purple arrowheads) are indicated. Shown are the results of one representative northern blot analysis (n = 3). The full blots, the loading controls, the RNA end profile with the detected 5′ and/or 3′ *rny*_ends obtained by RNA sequencing, and the trimming start and stop positions are shown in Supplementary Fig. 7 and the source data are provided as a Source Data file.

**Regulation of operon expression by RNase Y and PNPase.** We examined the impact of RNase Y and PNPase on operon expression by studying the *rsmC-cdd-bmpA* operon, described below (Fig. 9), and the *tsf-rpsB* operon (described in Supplementary Fig. 8), which was strongly upregulated in Δ*rny* (Supplementary Data 7).

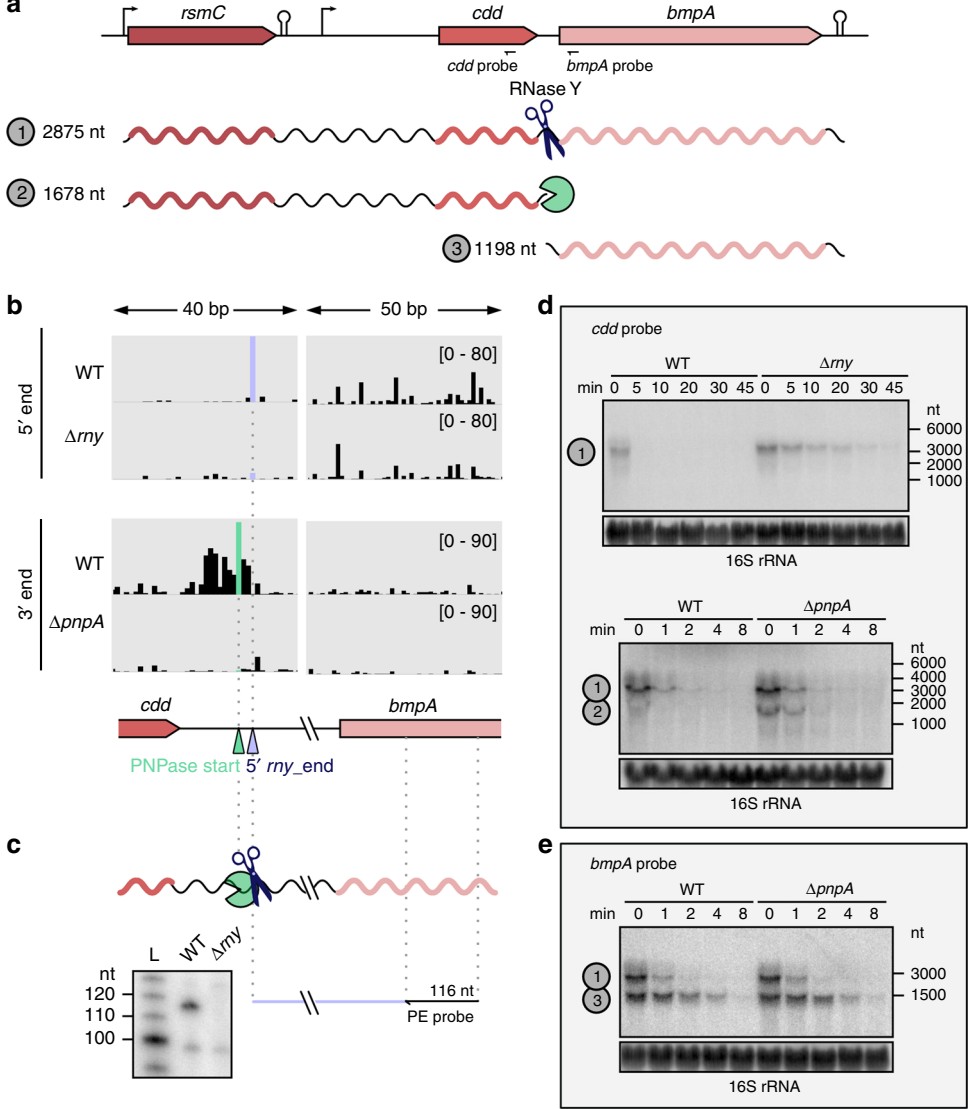

**Fig. 9 The concerted action of RNase Y and PNPase is responsible for the differential RNA stability of the *rsmC-cdd-bmpA* operon. a** Schematic representation of the *rsmC-cdd-bmpA* operon; the location of the promoter, terminator and probes used in the northern blot analyses and predicted RNA sizes are shown. **b** 5′ and 3′ end RNA sequencing coverages in the WT and Δ*rny* strains (for the 5′ end) and in the WT and Δ*pnpA* strains (for the 3′ end) of a region comprising portions of the *cdd* and *bmpA* ORFs and the intergenic region between the two genes. The coverage scale is indicated between brackets. The 5′ *rny*_end and the PNPase trimming start position identified in the *cdd-bmpA* intergenic region are depicted with purple and green arrowheads, respectively. **c** RNA 5′ end in the *cdd-bmpA* intergenic region in the WT and Δ*rny* strains, generated by RNase Y (scissors) and identified by primer extension analysis. The primer used is depicted with an arrow and binds upstream of the RNase Y processing and PNPase start positions ('pacman' symbol). The size of the expected cDNA product is indicated. Shown are the results of one representative primer extension experiment (*n* = 3). **d, e** The stability of *rsmC-bmpA_cdd*, *cdd* and *bmpA* RNAs was determined by northern blot analyses up to 45 or 8 min after the addition of rifampicin in the WT and Δ*rny* strains or in the WT and Δ*pnpA* strains. Shown are the results of one representative northern blot analysis (*n* = 3). The 16 rRNA was used as a loading control. Source data are provided as a Source Data file.

Based on the comparative analysis, we concluded that the *rsmC-cdd-bmpA* operon was targeted by both RNase Y and PNPase (Fig. 9a). This operon encodes a 16 rRNA methyltransferase (*rsmC*), a cytidine deaminase involved in pyrimidine metabolism (*cdd*) and a lipoprotein (*bmpA*) (Fig. 9a). RNase Y processed the transcript between *cdd* and *bmpA*, and a PNPase trimming start position was located a few nucleotides upstream of the 5′ *rny*_end (Fig. 9b). This observation indicates that the upstream fragment, corresponding to the *cdd* and *rsmC* ORFs, is subjected to PNPase degradation (Fig. 9b, Supplementary Data 5).

To establish the impact of RNase Y and PNPase activity, we assessed the stability of the different transcript isoforms of the operon by northern blot analyses (Fig. 9d). The stability of the full-length *rsmC-cdd-bmpA* transcript (~2900 nt) was greatly increased in the Δ*rny* strain (Fig. 9d). The *rsmC-cdd* RNA isoform (~1700 nt), which was barely detectable in the WT, was stabilized in the Δ*pnpA* strain (Fig. 9d). This result suggests that the *rsmC-cdd* isoform, arising from RNase Y processing, is rapidly degraded by PNPase. The *bmpA* isoform appeared to be more stable than the *rsmC-cdd* isoform in the WT strain, and the stability of this RNA was not affected by PNPase (Fig. 9e). In summary, the sequential activity of RNase Y and PNPase in the *cdd-bmpA* intergenic region ensures differential stability of the *rsmC-cdd* and *bmpA* RNAs.

## Discussion

We have investigated the targetome of RNase Y in the human bacterial pathogen *S. pyogenes*, using a method based on sequencing analysis of RNA 5′ and 3′ ends. We observed that the identified RNA 5′ and 3′ ends harboured distinct features in terms of sequence and structure conservation. Therefore, to further explore the origin of the RNase Y-dependent RNA ends, we developed an RNA-seq based comparative approach allowing us to juxtapose those data with 3′-to-5′ exoRNase targetomes. This method enabled us to determine that the detected RNA 5′ ends generated by RNase Y were usually not further trimmed. The 3′ ends, depending on RNase Y, resulted mostly from PNPase trimming and YhaM nibbling following RNase Y processing.

The analysis of the RNA 5′ ends generated by RNase Y revealed the presence of a G located just upstream of the processing sites for 87.4% of the targeted RNAs (Fig. 1f). The preference of RNase Y for this nucleotide at the processing site was first described in *S. aureus*, in which 58% of the processing sites were identified to be located upstream of a G[10]. A recent study from our laboratory demonstrated that RNase Y also requires a G to process *speB* mRNA, encoding a major virulence factor in *S. pyogenes*[11]. In light of the RNase Y cleavage signature identified in this study, it is likely that the G is required for the processing of substrates other than *speB* mRNA. Interestingly, in *B. subtilis*, a preferred sequence for RNase Y cleavage was not reported. Instead, this enzyme was shown to depend on the presence of RNA secondary structures around the processing site, as exemplified by the processing of several riboswitches[9], but this observation was never validated genome wide[18]. Similarly, *S. aureus* RNase Y processes the *saePQRS* transcript only when a secondary structure is located 6 nt downstream of the cleavage site[12]. In our study, the analysis of the MFE did not reveal a secondary structure in proximity of the 190 RNA 5′ ends (Fig. 1f). However, we noticed at these positions an increase in the MFE, which is consistent with the fact that RNase Y cleaves in single-stranded regions.

Although we showed that RNase Y is involved in RNA decay, we believe that, due to the limited number of direct targets identified, RNase Y might not be the major initiator of mRNA decay in *S. pyogenes*. We report 320 processing positions (identified by 5′ and 3′ end sequencing), which is consistent with previous reports for the *S. aureus* and *B. subtilis* RNase Y proteins, describing ~100 processing positions (identified by 5′ end sequencing)[10,18]. In *S. aureus*, the limited impact of RNase Y on global transcript stability is consistent with the low number of detected direct targets[10]. In contrast, for RNase E, the major endoRNase initiating RNA decay in Gram-negative bacteria, ~22,000 processing positions were identified in *Salmonella enterica*[8]. A possible explanation for the high number of RNase E processing events detected in this bacterium is the absence of RNase J1, which is found mainly in Gram-positive bacteria[2,27] and performs degradation from the 5′ end of the RNAs. In the present study, it is likely that we underestimated the number of RNase Y processing sites. First, because the method used relies on the detection of at least one RNA end (5′ or 3′), we did not identify RNase Y processing events when both generated ends were subsequently degraded by exoRNases (Fig. 10a). Second, the parameters used were stringent. However, the small RNase Y targetome found here is consistent with the fact that RNase Y is not essential under standard growth conditions (Supplementary Fig. 4). Comparison of the RNase Y and 3′-to-5′ exoRNase targetomes revealed 127 additional RNase Y processing sites that could be identified only in the absence of PNPase (due to the detection of decay intermediates), thereby increasing the total number of RNase Y processing sites identified in this study to 447 (Fig. 7 and Supplementary Fig. 6). It is possible that additional RNase Y processing positions were not detected in the Δ*pnpA*

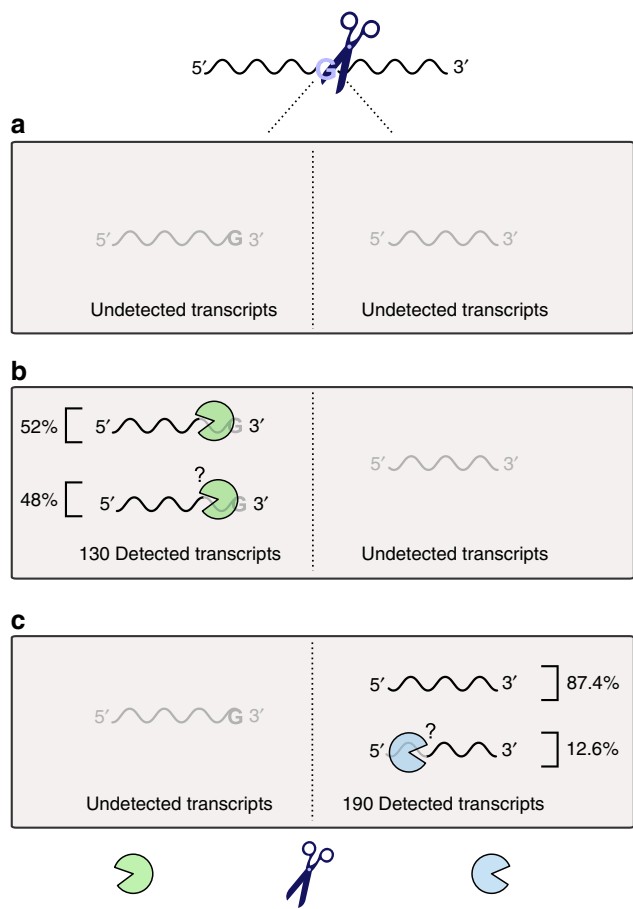

**Fig. 10 Fate of the RNAs cleaved by RNase Y in *S. pyogenes*.** RNase Y processing occurs preferentially after a G. The two processing products generated from the same molecule were never detected together. Subsequently to RNase Y activity, three different events are conceivable. **a** Both generated RNA fragments are degraded by exoRNases and/or endoRNases; hence, this activity would be undetectable in our experimental setting. **b** The RNA fragments upstream of the RNase Y processing position were detected (i.e., 3′ ends), but not the downstream products (i.e., 5′ ends). Among the 130 RNAs detected, we demonstrated that 52% of the RNA products were trimmed by 3′-to-5′ exoRNases (mainly PNPase and/ or YhaM). The remaining 48% were targeted either by several 3′-to-5′ exoRNases or by unidentified RNases. **c** The RNAs downstream of the RNase Y processing position were detected (i.e., 5′ ends), but not the upstream products (i.e., 3′ ends). Since 87.4% of the detected 5′ ends were mapped after G, we deduced that these RNAs were not further trimmed. The remaining 12.6% of the detected 5′ ends were not located after a G; therefore, we hypothesized that these ends were likely targeted by the 5′-to-3′ exoRNase J1.

strain due to functional redundancy between PNPase and RNase R[17]. Previously, a global RNA stability study in *S. pyogenes*, performed under conditions mimicking infection, revealed that deletion of *rny* causes the stabilization of 98% of the transcripts[28]. It would be interesting to characterize the RNase Y targetome under these conditions and to evaluate whether the increase in transcript stability correlates with RNase Y activity. Overall, RNA degradation in *S. pyogenes* must rely on another endoRNase(s) in addition to RNase Y. For example, the RNase J1/J2 complex could play an important role in RNA decay in *S. pyogenes*, as both enzymes are essential in this bacterium[5].

We demonstrated that RNase Y acts principally in concert with PNPase to degrade RNAs (Fig. 2; Supplementary Data 2 and 4). These two enzymes were shown to interact with each other in *B. subtilis*, although this interaction was not required for the degradation of all the studied targets[29]. When PNPase targeted the RNA fragments generated by RNase Y processing, we detected more PNPase trimming stop positions than start positions (Fig. 2). This result supports the observation that RNA 3′ ends produced by endoRNases are generally immediately degraded by PNPase and do not accumulate in the WT strain[16,30]. In our analysis, we could not detect RNase Y products entirely degraded by PNPase. Therefore, we suggest that the interplay between these two enzymes likely plays a broader role in RNA decay than that observed. By examining the PNPase targetome, we observed that some decay intermediates produced by RNase Y accumulated only in Δ*pnpA* (Fig. 7; Supplementary Fig. 6)[17]. Thus, the comparison of the Δ*pnpA* strain in the presence or absence of RNase Y led to the identification of additional RNase Y processing positions and a more representative picture of the interplay between these two enzymes.

Interestingly, we observed that the interplay of RNase Y and PNPase performs different functions in bacteria, such as decay of regulatory elements (e.g., riboswitches and T-boxes) and maturation of polycistronic mRNA. A role of RNase Y in the turnover of regulatory elements was previously observed in both *B. subtilis* and *S. aureus*[9,10,31]. Efficient removal of these regulatory elements from the bacteria might be important for the recycling of the ligand. In addition, RNase Y was previously shown to play an important role in the maturation of polycistronic transcripts by uncoupling the expression of genes encoded in the same operon[18,32]. Here, we show that the coordinated action of RNase Y processing in intergenic regions and subsequent degradation of one of the RNA products by PNPase results in differential decay of genes encoded within the same polycistronic mRNA, as exemplified for the *rsmC-cdd-bmpA* operon (Fig. 9).

As demonstrated previously, YhaM trims an average of 3 nt from most of the RNA 3′ ends generated by transcriptional terminators or by endoRNases[17]. Therefore, it was expected that YhaM would nibble the RNA 3′ ends produced by RNase Y. This activity complicated the identification of the original processing positions of RNase Y, as the G characterizing RNase Y activity was removed from the RNA 3′ end by YhaM (Supplementary Data 2 and 5). The example of SPy_0316 mRNA degradation illustrates that YhaM also targets RNAs already trimmed by other 3′-to-5′ exoRNases (Fig. 3d and Supplementary Fig. 2).

A previous study in *B. subtilis* suggested that YhaM could shorten the single-stranded RNA tail necessary for the binding of PNPase and RNase R to their targets, thereby protecting the RNAs from degradation by these two enzymes[16]. The observation that the three RNA fragments analysed in this study (Supplementary Fig. 5e) were not present in the Δ*yhaM* strain suggests that this hypothesis could also be valid, at least in a few cases, in *S. pyogenes*.

The number of RNase R trimming positions detected in *S. pyogenes* was limited during exponential growth in rich medium[17]; therefore, it was expected that the interplay between RNase Y and RNase R would also be restricted under these conditions. It is possible that RNase R and RNase Y might act in concert in different conditions than the ones tested.

Here, we focused on the fate of the RNA 3′ ends generated by RNase Y, and highlighted that these ends were in most cases further trimmed by the 3′-to-5′ exoRNases (Fig. 10b). In contrast to the detected RNA 3′ ends, the fate of the corresponding transcript 5′ ends is unknown. RNase J1, which degrades the RNAs in the 5′-to-3′ direction[6], might be a possible candidate enzyme for the degradation of these RNAs. Alternatively, these undetected

RNAs could be degraded by the 3′-to-5′ exoRNases up to the 5′ end produced by RNase Y, as exemplified by PNPase trimming of decay intermediates (Fig. 7 and Supplementary Fig. 6). In contrast to what we observed for the detected 3′ ends, 87.4% of the RNA 5′ ends corresponded to the original RNase Y processing positions (Fig. 10c). The remaining 12.6% of the RNA 5′ ends produced by RNase Y that were not located downstream of a G might correspond to RNase J1 trimming stop positions or might be generated by an endoRNase affected by RNase Y (Fig. 10c). The conditions leading to RNA 5′ end protection (detected 5′ ends) or degradation (undetected 5′ ends) have yet to be investigated. We believe that our method might facilitate the investigation of the concerted action of RNase Y and RNase J1, for instance, in *Streptococcus mutans*, in which none of these enzymes are essential.

To conclude, we have developed an RNA-seq based comparative approach that allows the genome-wide characterization of the specific RNase interplay and RNA degradation in vivo. We anticipate that this methodology will enable to elucidate diverse, parallel and interconnected, regulatory processes at the RNA level.

## Methods

**Bacterial culture.** *S. pyogenes* SF370 (M1GAS) and isogenic gene deletion strains (Supplementary Table 1) were grown in THY medium (Todd Hewitt Broth (THB, Bacto, Becton Dickinson) complemented with 0.2% yeast extract (Servabacter)) at 37 °C with 5% $CO_2$ without shaking[19]. TSA (trypticase soy agar, BD Difco) supplemented with 3% sheep blood was used as a solid medium.

**Growth curves.** Bacterial cultures were grown overnight (37 °C, 5% $CO_2$), diluted to an $OD_{620\,nm}$ of 0.02 in 5 ml of medium and centrifuged at $3200 \times g$ for 5 min. The resuspended pellet was used to inoculate flasks containing 25 ml of THY supplemented with 0.2% yeast extract. The growth was monitored by measuring the $OD_{620\,nm}$ using a microplate reader (Biotek PowerWave XS2). Growth curve experiments were performed in triplicate and the standard error of the mean was calculated.

**RNA isolation.** The overnight *S. pyogenes* cultures were diluted 1:200 into 300 ml of fresh THY medium and grown to an $OD_{620\,nm}$ of 0.25 corresponding to the mid-logarithmic growth phase. For the RNA stability assays, 250 µg/ml of rifampicin (Sigma-Aldrich) was added when the bacteria reached the mid-logarithmic phase of growth, and samples were taken either after 0, 5, 10, 20, 30 and 45 min or after 0, 1, 2, 4 and 8 min. The cells were rapidly harvested by mixing the 25 ml of cultures with 25 ml of 1:1 ice-cold acetone/ethanol solution and by centrifugation ($3500 \times g$ for 10 min at 4 °C). The pellets were thoroughly resuspended in 5 ml of TE buffer (50 mM Tris-HCl pH 8.0, 10 mM EDTA, pH 8, 50 mM NaCl, 25% sucrose). The cells were lysed by adding 100 µl of lysis buffer (20 mM Tris-HCl pH 8.0, 50 mM EDTA pH 8.0, 20% sucrose) supplemented with 2.5 mg/ml of lysozyme and 0.5 µg/µl of mutanolysin and incubated for 5 min on ice. The samples were mixed with the lysis executioner buffer (2% sodium dodecyl sulfate (SDS), 1 mg/ml Proteinase K) incubated at 95 °C for 1.5 min. Seven hundred and fifty microlitres of TRIzol reagent (Life Technologies) were added to the samples, which were subsequently inverted three times. After incubation for 5 min at room temperature, 200 µl of chloroform were added and the samples were mixed by vortexing. The samples were then incubated at room temperature for 10 min prior centrifugation at $11,300 \times g$ for 10 min at 4 °C. The upper aqueous phase (~700 µl) was gently collected and the RNAs were precipitated with ice-cold 100% isopropanol at a 1:1 ratio at −20 °C for at least 1 h. After centrifugation at $11,300 \times g$ for 10 min at 4 °C, the RNA pellets were washed with 1 ml of 70% ethanol, air dried for 10 min and dissolved in autoclaved Milli-Q $H_2O$. RNA integrity was assessed on 1% agarose gels.

**RNA sequencing and analysis.** The RNA sequencing was performed in biological triplicates using the workflow previously published by our laboratory[17]. After treatment with TURBO DNase (Ambion), the RNA quality was assessed using a bioanalyzer system (Agilent 2100). Subsequently, 4.5 µg of RNA was depleted of rRNAs (Ribo-Zero rRNA Removal Kit (Bacteria)) and treated with 10 U of RppH (New England Biolabs) at 37 °C for 1 h 30 min to convert the 5′ triphosphate RNAs in 5′ monophosphate RNAs. The RNAs were purified using standard extraction with phenol:chloroform:isoamylalcohol (25:24:1, Roth) and precipitated using ice-cold ethanol. The obtained RNAs were treated with T4 polynucleotide kinase (Thermo Scientific) according to the manufacturer's instructions to allow the subsequent ligation of the sequencing adaptors. After a purification step using the RNA Clean & Concentrator kit (Zymo Research), the RNAs were fragmented (Covaris M220) in a microTUBE AFA Fiber Pre-Slit Snap-Cap tubes for 140 s.

cDNA libraries were prepared using the NEXTflex® Small RNA Sequencing Kit v3 (Bioo Scientific) according to the manufacturer's instructions until step G. The purification step was performed using Agencourt AMPure XP beads (Beckman Coulter). The cDNA libraries were sequenced on a HiSeq3000 (paired-end mode, 75 bp) at the Max Planck-Genome-centre Cologne. The data were deposited in the National Center for Biotechnology Information (NCBI) sequence read archive (SRP149896). The numbers of sequencing reads obtained are listed in Supplementary Table 2. After quality filter (FastQC, v0.11.5) and adapter removal (Cutadapt v1.11), the reads were mapped to the *S. pyogenes* reference genome (NC_002737.2) and the total and end-specific (5′ and 3′) coverage profiles were visualized using the Integrative Genomics Viewer (IGV)[33,34]. Differentially expressed (DE) genes were identified using featureCounts (v1.5.2)[35] and edgeR (v3.20.6)[36,37] with absolute log2-fold change (log2 FC) ≥ 1 and false discovery rate (FDR) < 0.05.

**RNase Y processing sites**. RNase cleavage positions were identified following the previously published procedure[17]. In brief, the genome coverage data was pre-filtered with a count per million (cpm) value ≥0.05 and only the RNA ends displaying a cpm ≥5 were further analysed. We carried out differential expression analysis of normalized 5′ and 3′ read ends by comparing the following datasets (triplicates): WT vs. Δ*rny* and Δ*rny* vs. Δ*rny::rny*. RNase Y ends were identified using edgeR (v3.20.6)[36,37] with absolute log2 FC ≥ 1 and FDR < 0.05, and kept only if present in both comparisons. These identified RNA ends were named 5′ or 3′ *rny*_ends (Supplementary Data 1). These results were further filtered with additional parameters (i.e., the "proportion of ends" and the "ratio of WT and Δ*rny* proportion of ends") that were previously described[17]. First, the "proportion of end parameter", proportions of RNA ends at one position over the total RNA abundance, from the WT strain was set as ≥2%. Second, the "ratio of WT and Δ*rny* proportion of ends", WT proportion of ends over the Δ*rny* proportion of end, was set as ≥3%. This parameter allowed that the identification of RNA ends depending of an RNase Y processing event was independent from the RNA abundance in the WT and Δ*rny* strains.

**Comparison of the RNase Y and 3′-to-5′ exoRNase targetomes**. The RNase Y targetome (i.e., 5′ and 3′ *rny*_ends) was compared to the PNPase, YhaM and RNase R targetomes (i.e., 3′-to-5′ exoRNase trimming start and stop positions), which were previously identified (SRP149886)[17] (see Supplementary Fig. 1). Different approaches were used to perform the comparison. When at least two consecutive positions were identified as 3′ *rny*_ends in a window of 5 nt, the position with the highest ratio of proportion of ends between the WT strain and the Δ*rny* strain was selected for further analysis[17]. First, the 3′ *rny*_ends were compared with the PNPase, YhaM and RNase R trimming stop positions that were located 5 nt upstream or 5 nt downstream of the 3′ *rny*_ends (+/−5 nt shift) (Supplementary Data 2). Second, the 3′ *rny*_ends corresponding to 3′-to-5′ exoRNase trimming stop positions (in Supplementary Data 2) were compared with the trimming start positions located downstream (Supplementary Data 3). The maximum distance between these trimming stop and start positions was set at 200 nt for PNPase and RNase R, and 10 nt for YhaM. For RNase R, by setting a maximum distance of 200 nt, we did not identify any trimming start position downstream of the 3′ *rny*_ends that matched the RNase R stop positions. Third, the 3′ *rny*_ends were compared to the PNPase, YhaM and RNase R trimming start positions, allowing a +/−5 nt shift (Supplementary Data 4). Finally, the 5′ *rny*_ends were paired to the PNPase, YhaM and RNase R trimming start positions located 10 nt upstream (Supplementary Data 5). In addition, the 3′ *rny*_ends and 5′ *rny*_ends were compared by setting minimum and maximum distances of 40 and 1000 nt, respectively, between the ends (Supplementary Data 6). This comparison allowed the identification of RNA fragments produced by RNase Y. Python (v3.6.3) was used to perform all the comparisons described.

**Sequence logo and folding**. RNAfold (v2.4.3)[38] was used to calculate the MFE (ΔG in kcal/mol) using a sliding window of 50 nt sequences, with 100 or 200 nt centred on the position of interest. The average MFE at each nucleotide was then calculated. WebLogolib (v3.5.0) was used to generate the sequence logos[39], with sequences of 20 nt centred on the processing site with a GC content of 38.5%. The plots were generated using Python (v3.6.3) and matplotlib (v2.1.0).

**Northern blotting assays**. For the short RNAs, 10 μg of RNA was separated on 8% or 10% polyacrylamide/8 M urea gels (Figs. 5, 6 and 7; Supplementary Figs. 5, 6, and 7) in 1X TBE. The RNAs were transferred onto nylon membranes (HybondTM N+, GE Healthcare) with the Biorad Trans-Blot Cell system during 1 h and 15 min at 50 V in 1X TBE. RNAs were UV-crosslinked to the membranes with a Stratagene Stratalinker 1800 (two times in "Autocrosslink" mode). The oligonucleotide probes (40 pmol), listed in Supplementary Table 1 were [32]P labelled using T4 Poly-nucleotide Kinase (T4 PNK Fermentas) in presence of 2 μl of T4 PNK buffer (10×) and 2 μl of gamma-[32]P ATP (0.75 MBq, Hartmann analytic)[40]. The labelled oligonucleotides probes were subsequently purified over G-25 columns (GE Healthcare) following the manufacturer's instructions. The membranes were prehybridized in the Rapid-hyb buffer (GE Healthcare) for 1 h at 42 °C. The oligonucleotides were denatured and added to the membranes for an overnight

hybridization at 42 °C. The membranes were washed first with 5X SSC-0.1% SDS buffer and then with 1X SSC-0.1% SDS buffer for 15 min at 42 °C, respectively. RNA sizes were estimated using the RNA DecadeTM Marker (Ambion) or the ΦX174 DNA/HinfI Marker (Fermentas).

For the long RNAs (Fig. 8), 20 μg of RNA was separated on a 1% agarose gel (1X MOPS (20 mM MOPS free acid, 5 mM sodium acetate, 1 mM EDTA, pH 7.0), 6.6% formaldehyde) in 1X MOPS buffer with 0.7% formaldehyde for 2 h at 80 V. The separated RNAs were transferred onto a Nylon Hybond N + membrane (GE Healthcare) using a capillarity system overnight at room temperature in 20X SSC[11]. The RNAs were crosslinked to the membranes using UV (2X autocrosslinking, UV Stratalinker 1800). The membranes were pre-hybridized in Rapid-hyb buffer for 1 h (GE Healthcare) and subsequently incubated with denatured probes, which were [32]P labelled as described above. The hybridization was conducted overnight at 50 °C. The membranes were rinsed twice with pre-warmed 1X SSC + 0.1% SDS and subsequently with pre-warmed 0.5X SSC + 0.1% SDS. All washes were performed for 20 min at 50 °C. Long RNA sizes was estimated using the RiboRuler High Range Ladder (Thermo Scientific). A Typhoon Fla 9500 phosphorimager (Fujifilm) was used to visualize the radioactive signal for all northern blotting assays. As a loading control, 5S or 16S rRNAs were probed on the same membranes. Each northern blot was performed in at least triplicate. The uncropped blots are supplied in the Source Data file.

**Primer extension**. Primer extension was conducted on 10 μg of total RNA in biological triplicates[19]. RNAs were annealed to the radiolabelled primer (Supplementary Table 1) for 30 min at 65 °C and subsequently incubated on ice for 1 min and the RNAs were reverse transcribed using 1U of SuperScript III Reverse Transcriptase (Invitrogen) in the presence of 1X first strand buffer (Invitrogen), 5 mM dithiothreitol (DTT) and 40 U of RNaseOUT Recombinant Ribonuclease Inhibitor (Invitrogen), for 1 h at 55 °C. After inhibition of the SuperScript III enzyme by incubation at 70 °C for 15 min, the cDNAs were precipitated with ice-cold 100% ethanol at –20 °C for 1 h and centrifuged at 20,000 × *g* for 4 °C for 10 min. The pellet was washed with ice-cold 70% ethanol at 20,000 × *g* for 4 °C for 10 min and resuspended in 5 μl of 2X RNA loading dye. The cDNA products were resolved on 10% polyacrylamide/8 M urea/TBE gels and the size of the products was estimated using the AFLP 30–300 bp ladder labelled according to the manufacturer's instructions.

**Reporting summary**. Further information on research design is available in the Nature Research Reporting Summary linked to this article.

## Data availability
RNA sequencing data have been deposited at the NCBI under the accession number SRP149896. The data generated and analysed in this study are available from the corresponding authors upon request. All datasets generated in this study are available within the paper. The source data underlying Figs. 6c, 7c, 8b, 9c–e, and Supplementary Figs. 4, 5c–e, 6, 7, and 8b–e are provided as Source Data files; the source data underlying Figs. 7b and 3f are published (https://doi.org/10.1073/pnas.1809663115) and are available at NCBI under the accession number SRP149887.

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

## Acknowledgements
We acknowledge Richard Reinhardt and Bruno Huettel from the Max Planck-Genome-centre Cologne (MP-GC) for RNA sequencing of the cDNA libraries. We thank Johan Reimegård and Estelle Proux-Wera from SciLifeLab (Science for Life Laboratory, Sweden), and Knut Finstermeier and Davide Chiarugi from the Charpentier group for RNA sequencing data analysis support. We acknowledge Ciarán Condon and Petra Dersch for helpful discussions. The authors are grateful to the members of the Charpentier group for constructive discussions and critical reading of the paper. This work was supported by the Max Planck Society [E.C.], the Max Planck Foundation [E.C.], the Göran Gustafsson Foundation [Göran Gustafsson Prize to E.C.], the Alexander von Humboldt Foundation [AvH research fellowship to T.T.R.], the Kempe Foundation [E.C.], Umeå University [Dnr: 223-2386-10] [E.C.] and the Swedish Research Council [K2013-57X-21436-04-3] [E.C.]

## Author contributions
L.B., A.-L.L., A.L.R. and E.C. designed the study; L.B., A.-L.L., A.L.R., T.T.R. and K.H. performed the experiments; L.B., A.-L.L., A.L.R., T.T.R. and R.A.-B. performed the data analysis; L.B., A.-L.L. and A.L.R. interpreted the data; A.L.R. and E.C. oversaw the project; L.B., A.-L.L., A.L.R. and E.C. wrote the paper. All the authors read, edited and approved the paper.

## Competing interests
The authors declare no competing interests.
