## [Peer Review File · Nature Communications]

Reviewers' comments:

Reviewer #1 (Remarks to the Author):

The manuscript of Broglia et al. describes mapping of RNase Y cleavage sites (called “processing sites”) in *S. pyogenes*, and relates these to the previously described sites of 3’ exonuclease trimming start and stop positions. This is no doubt an important paper; the interplay between endonuclease and exonuclease activities in mRNA decay has long been postulated, but few observed examples of this have been described at the nucleotide level, as this manuscript does. In addition, the manuscript addresses RNase Y target-site specificity, demonstrating a conserved presence of a G nucleotide at the cleavage site. Little has been published about transcriptome-wide targets of RNase Y, a major decay-initiating endonuclease in other organisms. While the content is comprehensive, the nature of the subject makes this a difficult manuscript to read. The extensive supplementary figures are, at times, crucial to understanding the text, and so it is a pity that so much valuable information is present only in a supplemental file. There is even an entire section on p. 12 that refers solely to supplemental figures. Should not “supplemental” material be just that: not required for understanding the main text, but containing additional information? For example, Fig. 5 has five instances of the same concept, the effect of RNase Y cleavage in a 5’ regulatory region. The body of the text could show one example of this and leave the others to the supplementary file. If it is necessary to save space, Fig. 6 and its related text could be removed, as it is not news that endonuclease cleavage initiates decay that is accomplished by 3’ exonucleases. I am also not convinced that Fig. 8 is necessary.

I had one general criticism regarding the use of the word “transcript” for RNA fragments. Traditionally, a “transcript” is an RNA that is the direct result of transcription by RNA polymerase. A processed product thereof is referred to as an “RNA,” “RNA fragment,” “processing product,” “decay intermediate,” and similar. It is jarring to read the word “transcript” describing an RNA fragment that is a decay intermediate.

Other comments

1. line 29: "most of the time" (singular)
2. line 89, end: should be endo- and exoRNase-mediated
3. line 133 and throughout: reference to strain names is inconsistent. For example, the pnpA knockout strain may be called “the *deltapnpA* strain,” “the *deltapnpA*,” or just “*deltapnpA*.” Should be consistent.
4. line 135: “in the next two paragraphs.” I think what is meant is “in the next two sections.”

5. lines 178 and 273 and 448: should be “in the absence” or “in the presence”
6. line 179: should be “trimmed a few nucleotides”
7. line 182: “to stop” should be “and stopped”
8. line 200: “as previously described in other bacteria” sounds like the studies cited were in vivo studies. References 20 and 23 were in vitro studies. Suggest this should say, “as previously described for PNPase from other bacteria.”
9. line 219: the quotation doesn’t appear in the text. Should be: Here, we write “fully degraded” by PNPase for simplification.
10. line 228: Use of the word “retrieved” here is confusing (cf. two lines down). Suggest delete this word.
11. lines 257 and 263: “Table 6” should be “Table S6.”
12. line 297: I think this should be “S6C”
13. line 336: “corresponding to the first PNPase trimming stop position” Not clear which is “the first” – it appears that both positions have reduced levels in the delta rny strain.
14. lines 360-361: “and only the processed fragment downstream of bmpA was detected in the WT...” Does the Northern blot in Fig. 7D not show the rsmC/cdd RNA also in the WT?
15. line 362: should be “was located a few”
16. line 416: “lack of exoRNases starting from the 5’ end” is awkward. Suggest “lack of exoRNases that degrade from the 5’ end of transcripts and that are found mainly in...”
17. lines 455-456: The authors propose that removal of regulatory 5’ elements might be important for recycling of ligand. It may be helpful to cite here just such an observation from the article by Deikus et al. (2004; PMID 14976255), whose title is, “Recycling of a regulatory protein by degradation of the RNA to which it binds.”
18. line 460: “different” should be “differential”
19. lines 488-489: “RNase J1, the sole 5’-to-3’ exoRNase known in bacteria.” RNase J2 of *B. subtilis* also has 5’-to-3’ exonucleolytic activity, albeit weak (Mathy et al. 2009; PMID 20025672), as does RNase AM of *E. coli* (Ghodge and Raushel 2015; PMID 25871919).
20. Fig. 5: bottom right, “length” is misspelled
21. Fig. 6C: Having RNA 3 on top of RNA 4 here and having them in reverse order in Fig. 6E is confusing. Better to put 4 on top of 3 in Fig. 6C.

22. Fig. 6 legend: “grey rectangulars” and “purple rectangular” should be “grey rectangles” and “purple rectangle.” Also, comma missing after “which is equal for lanes within the grey rectangles,”
23. Fig. 8: Should the numbers “52%” and “48%” not rather be “58%” and “42%,” as in Fig. 2?
24. Supplementary Fig. S2A: “initial” is misspelled
25. Supplementary Fig. S2A legend: “stopped before the transcript termini.” The use of plural “termini” is strange. Suggest, “stopped before the transcript 5’ terminus”
26. Supplementary Fig. S2H and E legend: Order should be reversed (E and H)
27. Supplementary Fig. S5B, left: should be “3’ rny_end not matching exoRNase stop”
28. Supplementary Fig. S5C-E: missing comma after “which is equal for lanes within the grey rectangles,”
29. Supplementary Fig. S6 title: “intermediate decay fragments” perhaps should be “decay intermediate fragments”
30. Supplementary Fig. S6 first two lines: pacman and scissors symbols reversed

Reviewer #2 (Remarks to the Author):

The work in this manuscript is a follow up study building on a previous report in in which the authors mapped the ‘targetomes’ of 3’ exonucleases in *S. Pyrogenes*. In this work, the authors have mapped the targetome of the endonuclease RNase Y and correlated these cleavages sites with previously identified targets of the 3’ exonucleases. The work shows that fragments upstream of the RNase Y cleavage site are degraded by the 3’ exonucleases whereas the downstream fragments are resistant to further degradation. In particular, the 3’ exonuclease PNPase has a major role in the degradation of the upstream fragments. Together with the previous study, the work gives a detailed picture of the role of RNase Y and the 3’ exonucleases in mRNA processing and degradation in *S. pyrogenes*. In addition, similar to studies of the RNase Y targetome in other bacteria, this work shows that the number of RNase Y cleavages sites is relatively small and suggests that RNase Y is a ‘specialized’ endoribonuclease dedicated to the processing and degradation of a limited number of transcripts.

The manuscript is very well prepared. The text is clear. The experimental work and its presentation is of high quality. I only have a few minor comments.

Line 178. In 'the' absence...

Lines 192-197. The writing is a bit awkward (In our previous publication... In our previous work... Here...). I suggest that the authors try to combine the first two sentences into a single sentence.

Line 206-207. Why 'most likely not'? The result seems clear. I would state directly that the result suggests that a proportion of the upstream RNase Y cleavage products are not trimmed by the exonucleases. Beyond that, the reason is not clear. It could be that the exonucleases act slowly, that something is protecting the end (alternate RNA structures, RNA binding proteins, etc.), or that it is an exonuclease pausing site (i.e. not due to RNase Y cleavage).

Line 384. This method enabled 'us'...

Lines 426-431. The 2-fold slowdown in mRNA degradation reported in reference 31 seems rather small. Why do the authors believe that this should result in a significant slowdown of growth? I am only aware of one study in which *E. coli* growth rate was correlated with mRNA decay rate (S. Sousa et al., 2001, *Mol Microbiol* 42, 86). In this study, an approximately 2-fold increase in functional mRNA stability resulted in a 2-fold slowdown in growth rate. Furthermore, it is not known if the change in growth rate is due to the global slowdown of mRNA decay or effects on the level and/or processing of specific transcripts involved in regulating growth.

Point-to-point response to the referees' comments

Manuscript # NCOMMS1927498T (Broglia, Lécrivain et al.)

Referee #1 (Remarks to the Author):

General comments

The manuscript of Broglia et al. describes mapping of RNase Y cleavage sites (called “processing sites”) in *S. pyogenes*, and relates these to the previously described sites of 3' exonuclease trimming start and stop positions. This is no doubt an important paper; the interplay between endonuclease and exonuclease activities in mRNA decay has long been postulated, but few observed examples of this have been described at the nucleotide level, as this manuscript does. In addition, the manuscript addresses RNase Y target-site specificity, demonstrating a conserved presence of a G nucleotide at the cleavage site. Little has been published about transcriptome-wide targets of RNase Y, a major decay-initiating endonuclease in other organisms.

We thank the reviewer for her/his positive comments and feedback on the importance of our work.

While the content is comprehensive, the nature of the subject makes this a difficult manuscript to read. The extensive supplementary figures are, at times, crucial to understanding the text, and so it is a pity that so much valuable information is present only in a supplemental file. There is even an entire section on p. 12 that refers solely to supplemental figures. Should not “supplemental” material be just that: not required for understanding the main text, but containing additional information? For example, Fig. 5 has five instances of the same concept, the effect of RNase Y cleavage in a 5' regulatory region. The body of the text could show one example of this and leave the others to the supplementary file. If it is necessary to save space, Fig. 6 and its related text could be removed, as it is not news that endonuclease cleavage initiates decay that is accomplished by 3' exonucleases. I am also not convinced that Fig. 8 is necessary.

We thank the reviewer for her/his valuable suggestions and have reorganized the manuscript according to the reviewer's suggestions. In detail:

- We have added the data that were previously described in the paragraph p. 12 “*RNase Y generates decay intermediate fragments*” (now p. 12, “*RNase Y produces decay intermediates degraded by PNPase*”) to the main figures (now Figure 7).

- We have reduced the text related to the effect of RNase Y cleavage in 5' regulatory regions. We kept the example of T-boxes *serS* and *thrS* in the main figure (now Figure 8), but moved the other examples, with Northern blot analyses, to the supplementary data (now Supplementary Figure 7).
- The previous version of Figure 6 (*rpsB-tsfl* operon) and the related text were moved to the supplementary data (now Supplementary Figure 8).
- We decided to keep the previous version of Figure 8 (now Figure 10), schematically representing the model of RNA degradation in *S. pyogenes*, as main figure, since we believe that this figure enables the reader to better follow the discussion section and to better understand the main conclusion of this manuscript.
- In light of the reviewer's general comment to provide more valuable information in the main figures, we transferred most of the data from the extensive Supplementary Figure 2 to the current main Figure 3. In addition, we added a new main figure (now Figure 4) for the paragraph p. 9 "*Pairing 3' my_ends and exoRNase trimming start positions*", which in the previous manuscript was only associated to the previous Supplementary Figure 3. We are confident that these changes will help to facilitate the understanding of the main text.

I had one general criticism regarding the use of the word "transcript" for RNA fragments. Traditionally, a "transcript" is an RNA that is the direct result of transcription by RNA polymerase. A processed product thereof is referred to as an "RNA," "RNA fragment," "processing product," "decay intermediate," and similar. It is jarring to read the word "transcript" describing an RNA fragment that is a decay intermediate.

We thank the reviewer for this observation. All sentences containing "transcript" when referring to RNA decay fragments were revised accordingly in the main and supplementary text and figures (see red text).

Other comments

1. line 29: "most of the time" (singular)
2. line 89, end: should be endo- and exoRNase-mediated
3. line 133 and throughout: reference to strain names is inconsistent. For example, the *pnpA* knockout strain may be called "the *deltapnpA* strain," "the *deltapnpA*," or just "*deltapnpA*." Should be consistent.
4. line 135: "in the next two paragraphs." I think what is meant is "in the next two sections."
5. lines 178 and 273 and 448: should be "in the absence" or "in the presence"
6. line 179: should be "trimmed a few nucleotides"

7. line 182: "to stop" should be "and stopped"
8. line 200: "as previously described in other bacteria" sounds like the studies cited were in vivo studies. References 20 and 23 were in vitro studies. Suggest this should say, "as previously described for PNPase from other bacteria."
9. line 219: the quotation doesn't appear in the text. Should be: Here, we write "fully degraded" by PNPase for simplification.
10. line 228: Use of the word "retrieved" here is confusing (cf. two lines down). Suggest delete this word.
11. lines 257 and 263: "Table 6" should be "Table S6."
12. line 297: I think this should be "S6C"
13. line 336: "corresponding to the first PNPase trimming stop position" Not clear which is "the first" – it appears that both positions have reduced levels in the delta my strain.
14. lines 360-361: "and only the processed fragment downstream of bmpA was detected in the WT..." Does the Northern blot in Fig. 7D not show the rsmC/cdd RNA also in the WT?
15. line 362: should be "was located a few"
16. line 416: "lack of exoRNases starting from the 5' end" is awkward. Suggest "lack of exoRNases that degrade from the 5' end of transcripts and that are found mainly in..."
17. lines 455-456: The authors propose that removal of regulatory 5' elements might be important for recycling of ligand. It may be helpful to cite here just such an observation from the article by Deikus et al. (2004; PMID 14976255), whose title is, "Recycling of a regulatory protein by degradation of the RNA to which it binds."
18. line 460: "different" should be "differential"
19. lines 488-489: "RNase J1, the sole 5'-to-3' exoRNase known in bacteria." RNase J2 of *B. subtilis* also has 5'-to-3' exonucleolytic activity, albeit weak (Mathy et al. 2009; PMID 20025672), as does RNase AM of *E. coli* (Ghodge and Raushel 2015; PMID 25871919).
20. Fig. 5: bottom right, "length" is misspelled
21. Fig. 6C: Having RNA 3 on top of RNA 4 here and having them in reverse order in Fig. 6E is confusing. Better to put 4 on top of 3 in Fig. 6C.
22. Fig. 6 legend: "grey rectangulars" and "purple rectangular" should be "grey rectangles" and "purple rectangle." Also, comma missing after "which is equal for lanes within the grey rectangles,"
23. Fig. 8: Should the numbers "52%" and "48%" not rather be "58%" and "42%," as in Fig. 2?
24. Supplementary Fig. S2A: "initial" is misspelled
25. Supplementary Fig. S2A legend: "stopped before the transcript termini." The use of plural "termini" is strange. Suggest, "stopped before the transcript 5' terminus"

26. Supplementary Fig. S2H and E legend: Order should be reversed (E and H)
27. Supplementary Fig. S5B, left: should be "3' rny_end not matching exoRNase stop"
28. Supplementary Fig. S5C-E: missing comma after "which is equal for lanes within the grey rectangles,"
29. Supplementary Fig. S6 title: "intermediate decay fragments" perhaps should be "decay intermediate fragments"
30. Supplementary Fig. S6 first two lines: pacman and scissors symbols reversed

We thank the reviewer for her/his suggestions to improve the manuscript. We have now revised all sentences and figures according to the comments (see red in the text and figure legends).

In particular:

13. We agree with the reviewer that the abundance of both PNPase trimming stop positions were reduced in Δrny and therefore both positions appear to be RNase Y-dependent. We have now revised the sentence (now in the legend of the Supplementary Figure 8d).

14. We agree with the reviewer that the sentence was not correct. We intended to say that the *rsmC-cdd* RNA was barely detectable in the WT strain because it was subject to rapid degradation by an exoRNase. We have now revised the text (p13; lines 310-312).

Referee #2 (Remarks to the Author):

General Comments

The work in this manuscript is a follow up study building on a previous report in which the authors mapped the 'targetomes' of 3' exonucleases in *S. Pyrogenes*. In this work, the authors have mapped the targetome of the endonuclease RNase Y and correlated these cleavages sites with previously identified targets of the 3' exonucleases. The work shows that fragments upstream of the RNase Y cleavage site are degraded by the 3' exonucleases whereas the downstream fragments are resistant to further degradation. In particular, the 3' exonuclease PNPase has a major role in the degradation of the upstream fragments. Together with the previous study, the work gives a detailed picture of the role of RNase Y and the 3' exonucleases in mRNA processing and degradation in *S. pyrogenes*. In addition, similar to studies of the RNase Y targetome in other bacteria, this work shows that the number of RNase Y cleavages sites is relatively small and suggests that RNase Y is a 'specialized' endoribonuclease dedicated to the processing and degradation of a limited number of transcripts. The manuscript is very well prepared. The text is clear. The experimental work and its presentation is of high quality. I only have a few minor comments.

We thank the reviewer for her/his positive comments and feedback on the quality and clarity of our work.

Other Comments

Line 178. In 'the' absence...

Lines 192-197. The writing is a bit awkward (In our previous publication... In our previous work... Here...). I suggest that the authors try of combine the first two sentences into a single sentence.

Line 206-207. Why 'most likely not'? The result seems clear. I would state directly that the result suggests that a proportion of the upstream RNase Y cleavage products are not trimmed by the exonucleases. Beyond that, the reason is not clear. It could be that the exonucleases act slowly, that something is protecting the end (alternate RNA structures, RNA binding proteins, etc.), or that it is an exonuclease pausing site (i.e. not due to RNase Y cleavage).

Line 384. This method enabled 'us'...

Lines 426-431. The 2-fold slowdown in mRNA degradation reported in reference 31 seems rather small. Why do the authors believe that this should result in a significant slowdown of growth? I am only aware of one study in which E. coli growth rate was correlated with mRNA decay rate (S. Sousa et al., 2001, Mol Microbiol 42, 86). In this study, an approximately 2-fold increase in functional mRNA stability resulted in a 2-fold slowdown in growth rate. Furthermore, it is not known if the change in growth rate is due to the global slowdown of mRNA decay or effects on the level and/or processing of specific transcripts involved in regulating growth.

We thank the reviewer for her/his observations and suggestions. We have now revised the sentences accordingly to the comments (see red in the text).

In particular:

Line 206-207. We agree with the reviewer that the comparison of the 3' *my_* ends with the 3'-to-5' *exoRNase* trimming start positions (p. 9 and now Figure 4) clearly shows that while a subset of transcript 3' ends generated by RNase Y are further degraded, another subset was not subject to degradation yet. We have therefore revised this part (p. 9; lines 193-196: *"Therefore, the 3' ends generated by RNase Y are targeted by these *exoRNases*. Their*

detection in this analysis suggests that a portion of the RNAs was not yet subjected to 3'-to-5' exoRNase degradation.”)

Lines 426-431. In the cited paper (reference 28, Chen *et al.*, 2013 J. Bacteriol. 195 (11) 2585-2594), the deletion of *rny* affected the stability of almost all cellular transcripts (98%). We do not believe that this effect should result in a significant reduction of bacterial growth. Instead, we implied that this global increase in transcript stability that was observed in the absence of RNase Y does not correlate with the limited number of RNase Y cleavage sites that we have identified. However, we believe that we cannot compare the RNase Y targetome, characterized here, with the global transcript stability study (reference 28) because, while we used a rich medium for the growth of *S. pyogenes*, the previous study was performed using a medium poor in carbohydrates, mimicking infection conditions. Therefore, we have now revised this part accordingly (now p. 17; lines 376-381: *Previously, a global RNA stability study in S. pyogenes, performed under conditions mimicking infection, revealed that deletion of rny causes the stabilization of 98% of the transcripts*²⁸. *It would be interesting to characterize the RNase Y targetome under these conditions and to evaluate whether the increased in transcript stability correlates with RNase Y activity.*).

REVIEWERS' COMMENTS:

Reviewer #1 (Remarks to the Author):

The new version of the manuscript of Broglia et al. has satisfactorily addressed reviewers' comments, incorporating almost all of the suggested revisions. The text has been clarified in many places and the figures reworked to make this manuscript more accessible to the reader.

David Bechhofer

Reviewer #2 (Remarks to the Author):

none